# UK and US risk factors for hearing loss in neonatal intensive care unit infants

**Sally K. Thornton** [1,2]*, **Derek J. Hoare**[1,2], **Alice M. Yates**[1], **Karen R. Willis**[3], **Polly Scutt**[1,2], **Padraig T. Kitterick**[1,2], **Abhijit Dixit**[4], **Dulip S. Jayasinghe**[5]

**1** Hearing Sciences, Mental Health and Clinical Neurosciences, School of Medicine, The University of Nottingham, Nottingham, United Kingdom, **2** NIHR Nottingham Biomedical Research Centre, Nottingham, United Kingdom, **3** The Children's Audiology, Nottingham University Hospitals, Nottingham, United Kingdom, **4** Clinical Genetics, Nottingham University Hospitals, Nottingham, United Kingdom, **5** Neonatal Intensive Care Unit, Nottingham University Hospitals, Nottingham, United Kingdom

\* sally.thornton@nottingham.ac.uk

**Data Availability Statement:** Participants did not give consent on data sharing, in accordance with the European General Data Protection Regulation and the competent ethic committees, only blinded

## Abstract

### Importance

Early detection and intervention of hearing loss may mitigate negative effects on children's development. Children who were admitted to the neonatal intensive care unit (NICU) as babies are particularly susceptible to hearing loss and risk factors are vital for surveillance.

### Design, setting and participants

This single-centre retrospective cohort study included data from 142 inborn infants who had been admitted to the NICU in a tertiary regional referral centre. Data were recorded for 71 infants with confirmed permanent congenital hearing loss hearing loss. To determine impact of NICU admission independently of prematurity, babies were individually matched with 71 inborn infants on gestational age, birthweight, and sex.

### Main outcomes and measures

Neonatal indicators were recorded for all children with permanent congenital hearing loss. Presence of UK and US risk factors for hearing loss were collected on the neonatal population with hearing loss and for the matched controls.

### Results

A fifth (21%) of babies with hearing loss had one or more UK risk factors whereas most (86%) had at least one US risk factor. False positives would be evident if US factors were used whereas the matched controls had no UK risk factors. Ten babies who at birth had no UK or US risk factors did not have any significant neonatal indicators identified in their records, one was ventilated for one day and two had a genetic anomaly.

### Conclusions and relevance

Current risk factors for hearing loss we identified for follow-up in this high-risk group are highly specific for congenital hearing loss. UK risk factors were highly specific for hearing

data are available. Contact information for requests on data sharing are the following: Dr Mahendra Amita, Amita.Mahendru@nuh.nhs.uk Queens Medical Centre Nottingham University Hospitals Tel: 0115 924 9924 ext.86470

**Funding:** The funders had no role in study design, data collection and analysis, decision to publish, or preparation of the manuscript. DJH and SKT are funded by the National Institute for Health and Care Research (NIHR) Biomedical Research Centre programme. The views expressed are those of the authors, and not necessarily those of the NIHR, the NHS, or the Department of Health and Social Care.

**Competing interests:** The authors have declared that no competing interests exist.

loss but not sensitive and conversely, US risk factors are sensitive but not specific so false positives would be recorded. A national study of neonatal indicators could provide the utility to test which combinations of risk factors provide high sensitivity without losing specificity.

## 1. Introduction

With ever advancing medical technology it is important to revisit hearing loss (HL) in neonatal intensive care unit (NICU) graduates. Incidence and prognosis according to some risk factors will likely change over time. Conversely, neonatal indicators for HL are important to explore in this high-risk group, as more than a third of countries do not have a newborn hearing screening programme (NHSP) [1].

To date there are few studies on the association between neonatal indicators and HL in high-risk populations [2–7]. No NICU studies have compared the risk factors currently in UK and US guidance. Many studies have a limited sample size, and not all studies have included babies with confirmed permanent HL. It may also prove to be useful to provide a matched control group on gestational age and weight as then risk factors can be determined independent of prematurity.

HL is one of the most common congenital anomalies, occurring in approximately 1–3 infants per 1,000, and is the fourth leading cause of disability globally [8–10]. There are serious negative consequences associated with undetected HL such as speech, language and cognitive delays, poor social adjustment, and academic achievement [11–13]. Physical and psychological morbidities may also co-exist with HL [14] and more than a quarter of children with HL have additional disabling conditions [15, 16]. An aetiologic factor can be identified in 50–60% of HL cases–a genetic cause is present in 60% of infants with HL, peripartum problems in 21%, and cytomegalovirus (CMV) infection in 19% [17–21].

Neonatal indicators are predictive of HL in newborn infants–where prematurity and admission to the neonatal department are the predictors of HL [22, 23]. The very preterm infants have a high risk of functional impairment of the brainstem auditory pathway [23]. Prevalence of HL increases with decreasing gestational age and decreasing birthweight [24]. Neonates born at a lower birth weight (<750g) showed a higher frequency of HL than other infants because they are more likely to have developed bronchodysplasia, and additional neurological morbidities [25]. Previous data indicate that other neonatal factors; e.g; mechanical ventilation [26] oxygen therapy [27] and hyperbilirubinemia [28] are also associated with HL. Recently there has been interest in co-morbidities of multiple risk factors for HL [14] and Chant et al. (2022) have shown cumulative doses of ototoxic medication are associated with increased risk of HL in preterm infants [7].

There is international consensus that early diagnosis and follow-up of HL in children is important for them to achieve satisfactory language outcomes [13, 29, 30]

Highly successful implementation of automated auditory brainstem response (AABR) hearing screening has been demonstrated in the NICU setting and there is good evidence that newborn hearing screening programmes (NHSP) achieve early identification of HL [31]. Regardless of their hearing screening result, many NICU babies meet at-risk criteria for referral and additional hearing testing at 7–9 months according to US (Joint committee on infant hearing [JCIH]) and UK (National Institute for Health and Care Excellence NHSP) guidelines [32, 33]. Table 1. summarises current UK and US risk factors for hearing loss.

Neonatal indicators for HL remain important to investigate in resource rich countries because they could change in-line with introduction of new technologies on the NICU. Many

**Table 1. UK and US risk factors for referral.**

| UK Risk factors for hearing loss in the neonatal period according to NHSP 2022 [33] | US Risk factors for hearing loss in the neonatal period according to JCIH 2019[32] |
|---|---|
| 1. Syndromes associated with hearing loss (including Down's syndrome)<br>2. Cranio-facial abnormalities, including cleft palate<br>3. Confirmed congenital infection (toxoplasmosis or rubella)<br>4. Been in a special care baby unit (SCBU) or neonatal intensive care unit (NICU) over 48 hours, with no clear response automated otoacoustic emissions (AOAE) test for both ears, but clear response automated auditory brainstem response (AABR) for both ears | 1. A history of family members being deaf or hard of hearing with onset in childhood<br>2. Infants who require care in the NICU or special care nursery for more than five days is used as an indicator of illness severity.<br>3. Hyperbilirubinemia<br>4. Aminoglycoside administration of > 5 days<br>5. Perinatal asphyxia, HIE particularly if requiring cooling treatment.<br>6. Extracorporeal membrane oxygenation (ECMO)<br>7. In-utero infections (i.e., herpes, rubella, syphilis, and toxoplasmosis). CMV and Zika also<br>8. All craniofacial conditions and physical conditions associated with hearing loss<br>9. More than 400 syndromes and genetic disorders associated with atypical hearing thresholds<br>10. Peri/postnatal risk factors<br> i. Perinatal and postnatal confirmed bacterial and/or viral meningitis or encephalitis (especially herpes viruses and varicella and hemophilus influenza and pneumococcal meningitis<br> ii. Head trauma particularly injury to the mastoid and chemotherapy<br> iii. Family/caregiver concern regarding hearing, speech, language, or development requires attention. |

countries do not have a NHSP and do not have the resources to automatically refer premature neonates for hearing screening; 38% of the world's newborns have minimal or no hearing screening [1].

There is a paucity of recent studies of HL in the NICU population, and many have not accounted for the confounding risk factors of low birth weight and prematurity. None have compared the risk factors currently in the UK and US guidance.

The aim of the present study was to determine the reliability of known risk factors in the prediction of HL in neonates.

### 1.1. Objectives

1) To determine the sensitivity and specificity for current UK and US risk factors for hearing loss in these infants. 2) Describe neonatal indicators for babies who would *not* be targeted by UK or US risk factors.

## 2. Materials and methods

This study used data from the Nottingham NICU research database (NEAT) under ethical approval to analyse routinely collected data (REC project ID: 292263), South Central Berkshire Research Ethics Committee. The data were accessed for research purposes from September 28th 2022 until March 22nd 2023 and members of the NEAT database team had access to information that could identify individual participants during data collection. The data was pseudonymised after data collection.

### 2.1 Description of HL cases and matched controls

The study population consisted of 14,037 babies born in Nottingham and admitted to the NICU in Nottingham between March 1st 2008 and February 29th 2020. The cases constitute 71

babies admitted to the NICU in this same epoch identified on NHSP and then followed up to confirm congenital permanent HL. When HL is cited in reference to the data in this study, we are referring to confirmed permanent congenital HL which includes bilateral HL and unilateral HL. We categorised HL in this study according to British Society of Audiology guidelines as mild, moderate, severe, or profound where hearing threshold in dB HL is averaged over the frequencies 0.5, 1, 2, and 4 kHz for the better hearing ear (www.thebsa.org.uk). We had a range of ages of children (birth dates from March 1st 2008 and February 29th 2020) and the degree of HL was taken from their most recent audiological report. The hearing data were taken from the pure-tone audiogram in the main, but also play and visual reinforcement audiometry depending on age and stage of development. The hearing data extracted from air and bone conduction tests (diagnostic ABR) indicated that 59 children had sensorineural hearing loss (17 with auditory neuropathy spectrum disorder), 7 had conductive hearing loss and 5 had mixed hearing loss.

Of the 71 cases of HL, 46 babies had confirmed permanent congenital BHL and 25 had confirmed congenital permanent UHL. Eight children had mild BHL (20–40 dB HL), 24 had moderate BHL (41–70 dB HL), four had severe BHL (71-95dB HL) and eight had profound BHL >95 dB HL. Fourteen of the patients with UHL had mild or moderate hearing loss (one mild, 13 moderate) and 10 had severe and two had profound hearing loss in their affected ear. All infants with HL were matched on gestational age (±1 week), gestational weight (±100g), sex, and whether they were on the NICU for ≤ or ≥48 hours. They were matched with infants who passed NHSP and additionally had no known acquired HL at the time of data collection. Children were matched on these factors as it is already established in the literature that gestational age and weight impact risk of hearing loss and we wanted to study the neonatal indicators independent of the degree of prematurity.

We collected the neonatal indicators for HL from routinely recorded data on the neonatal clinical database Badgernet (Clevermed, Edinburgh) from paper notes and other local clinical databases. These neonatal indicators included all UK and US risk factors (Table 1).

**2.1.1 Inclusion criteria.** Inborn neonates admitted on the day of birth to Nottingham NICU March 1st, 2008- February 29th, 2020. The HL cases had to have permanent congenital HL (referred on NHSP and confirmed with later audiological testing). There was also a requirement that they had their neonatal indicators input onto Badgernet the neonatal database.

**2.1.2 Exclusion criteria.** Matched control cases were excluded if they were referred on the NHSP or if they later went on to have an acquired HL (at the time of data collection). Cases were also excluded if they had their hearing screening out of region.

## 2.2 Data entry and items

Data entry onto NICU Badgernet database was completed by a member of the NICU medical team. A confirmed in utero infection includes confirmed viral infections, (CMV, rubella etc). Individual cases were screened by a Consultant Neonatologist to indicate that these babies had a confirmed viral infection. Confirmed meningitis was an indication in the notes that the baby had a positive septic screen and cerebral spinal fluid parameters indicating that they had any type of meningitis (bacterial or viral). Suspected CMV cases were managed according to local guidelines, no national guidelines are or were in place at the time of data collection [local: NUHT Clinical Guidelines and Policies catalogue (koha-ptfs.co.uk)].

Early-onset infection included any mention of infection (bacterial or viral) in the first 72 hours recorded as part of their principal diagnoses. The data on trough/peak serum concentrations of antibiotics, the number of doses administered and duration it was administered was

not routinely recorded on Badgernet. For this study analyses of ototoxic medication were not possible, and we only recorded binary data i.e., if they were given one or more doses of an aminoglycoside antibiotic, usually gentamicin. Total days on the NICU were calculated from subtracting birth date from discharge date form the NICU. There are several risk factors which are observed in the well-baby population and are risk factors for HL but are not routinely observed in infants admitted to the NICU; these included chemotherapy, neurodegenerative disorders and temporal bone skull fracture. There were no cases recorded of any of these risk factors in either HL cases or matched controls during the 12 years of data from this large tertiary centre.

### 2.3 Statistical analyses

A power calculation was not performed as the sample size was opportunistic and limited by the number of available HL cases with NICU admission at the Nottingham University Hospitals. Data were analysed with SPSS version 28.0 (SPSS Inc. Chicago, H.). A medical statistician independently analysed the data for this study.

The data were normally distributed but skewed sufficiently to indicate non-parametric statistical tests should be used. In the subgroup analyses, neonatal indicators for HL cases were compared with their own matched controls. The data analysed include the number of UK and US risk factors. The "At least one UK risk factor (%)" and equivalent US rows are a binary version of the variable, the p-value comes from a chi-squared test. Binary logistic regression was used to assess whether current UK and US risk factors are associated with hearing loss in the NICU population. We looked at each risk factor individually and found the sensitivity, specificity and PPV of each risk factor on its own and identified the relationship between risk factor and HL to give an OR. We used this to determine whether any of the UK or US risk factors were statistically significantly associated with HL.

## 3. Results

A total of 14,037 neonates were admitted to the NICU in Nottingham over 12 years. Of those, 6,246 were admitted to the NICU and stayed on the NICU ≥48 hours.

Mean gestational age for all babies (HL cases and controls) was 34.6 weeks and mean birth weight was 1890 g. Incidence of permanent HL was 0.5% of which n = 46 was confirmed permanent congenital BHL and n = 25 confirmed permanent congenital UHL.

One-hundred and forty-two neonates were included in these analyses: 71 with permanent congenital hearing loss and 71 matched case-controls with no HL.

### 3.1 Comparison of UK and US risk factors

UK risk factors for HL were specific but not sensitive for hearing loss. Table 2 indicates that most HL babies (56/71, 79%) had no UK risk factors and 14% (10/71) had no US risk factors. However, UK risk factors are specific compared with US risk factors as no controls had any (≥1) of the UK risk factors, whereas the majority (66%, 47/71) of controls had at least one US risk factor. Including a stay on the NICU ≥ 48 hours as a UK risk factor, increased sensitivity to 92% for babies with HL but 80% of matched controls would also be targeted.

Using binary logistic regression there was evidence that the proportions of HL cases/ matched controls changed as the number of risk factors increase; this was true for both UK and US risk factors (Cochrane-Armitage test; P = .0001, UK risk factors; P = 0.003 US risk factors).

Table 3 indicates that current UK risk factors and many US risk factors had high specificity (and PPV) and low sensitivity, excepting HIE and NICU stay >5 days which have high

**Table 2. UK and US risk factors for babies with HL.**

|  | HL cases | Matched controls | p-value |
| --- | --- | --- | --- |
|  | n = 71 | n = 71 |  |
| **Number of UK risk factors** |  |  |  |
| 0 | 56 (79%) | 71 (100%) |  |
| 1 | 11 (15%) | 0 (0%) |  |
| 2 | 4 (6%) | 0 (0%) |  |
| At least 1 UK risk factor, n (%) | 15 (21%) | 0 (0%) | 0.000042 |
| **Number of US risk factors** |  |  |  |
| 0 | 10 (14%) | 24 (34%) |  |
| 1 | 5 (7%) | 3 (4%) |  |
| 2 | 40 (56%) | 36 (51%) |  |
| 3 | 13 (18%) | 8 (11%) |  |
| 4 | 3 (4%) | 0 (0%) |  |
| At least 1 US risk factor, n (%) | 61 (86%) | 47 (66%) | 0.0059 |

Stay on the NICU>48 hours are not included in the UK risk factors as this is one of the factors that was matched on.

sensitivity and low specificity. High PPV results can be interpreted as indicating the accuracy of the statistics and show how likely it is a patient with the risk factor has hearing loss.

## 3.2 Children with HL who have *no* US or UK risk factors

There are 10/71 (14%) babies that have a HL who would not have been targeted for HL follow-up by any of the UK or US risk factors (excluding days on the NICU) if they had not been referred from NHSP. All these babies were born near term (≥36 weeks) weighing >2kg. Only one baby was ventilated for one day and they had no other neonatal indicators discernible from patient records. One baby with UHL was on the NICU > 5days (would be identified by US risk factor) and two other babies were admitted for more than 24 hours. Six babies had genetic testing, and two babies tested positive for changes in connexin 26. In the BHL group,

**Table 3. Current UK and US risk factors for HL–binary logistic regression.**

| Risk factors | Sensitivity | Specificity | PPV | p-value |
| --- | --- | --- | --- | --- |
| **Risk factor (UK)** |  |  |  |  |
| Cranio-facial anomaly | 0.18 | 1.00 | 1.00 | 0.97 |
| Syndrome associated hearing loss | 0.07 | 1.00 | 1.00 | 0.98 |
| In-utero infection | 0.01 | 1.00 | 1.00 | 0.99 |
| **Risk factor (US)** |  |  |  |  |
| Cranio-facial anomaly | 0.18 | 1.00 | 1.00 | 0.97 |
| Syndrome associated hearing loss | 0.07 | 1.00 | 1.00 | 0.98 |
| In-utero infection | 0.01 | 1.00 | 1.00 | 0.99 |
| HIE | 0.94 | 0.09 | 0.51 | 0.35 |
| Meningitis | 0.03 | 0.99 | 0.67 | 0.57 |
| Exchange transfusion | 0.01 | 1.00 | 1.00 | 0.99 |
| NICU stay > 5 days | 0.79 | 0.34 | 0.54 | 0.093 |

NICU = neonatal intensive care unit, PPV = positive predictive value. NICU stay > 48 hours has not been included as this was one of the criteria the data were matched on (NICU stay ≥48 or ≤48 hours).

one child with BHL had gross motor delay and two BHL cases were diagnosed later with autistic spectrum disorder. Imaging data were only available for two of these children where one had a normal MRI and the other one had dilatation of vestibular aqueducts.

Intrauterine growth restriction was listed as a principal diagnosis for two babies out the ten who had no UK/US risk factors (one with UHL and one with BHL). Data for head circumference measurement was only available for one baby with BHL (34cm), all four babies with UHL had head circumference measures (31, 37.5, 31 and 32cm). Average head circumferences for matched babies were 30cm (BHL) and 33cm (UHL).

### 3.3 Genetics

For children with UHL there was genetic testing in 50% (12/25) of cases totalling 17 tests in 12 children. Ten tests were normal, four had pathogenic variants and two were variants of unknown significance (VUS), but none were causally related to deafness. For the babies with BHL there was genetic testing in 70% (32/46) of children and a total of 69 tests in the 32 children. Fifty-two (52/62) tests were normal. Four identified pathogenic variants causative of deafness, three were VUS, and three identified carrier statuses of recessive conditions.

## 4 Discussion

The aim of the present study was to investigate the official UK and US risk factors from the neonatal database in a large tertiary referral centre. These data are important as risk factors change over time due to technological advancements on the NICU, furthermore, risk factors are used in countries where NHSP are not in place for targeting follow-up for children with HL.

It has been shown that the probability of HL increases with an increase in the number of risk factors [34]. Most (86%) of our HL cases had at least one US risk factor and in contrast only 21% of HL cases had one or more UK risk factors. Many of the individual risk factors we recorded had a high degree of specificity (and PPV) but low sensitivity. The downside of using US risk factors were that they were not specific, and many (66%) matched controls had one or more factors, whereas no matched controls had more than one UK risk factor.

The current report is distinct from some other studies on HL in high-risk populations because the babies in this study have confirmed permanent HL. We provide a matched population for comparison on birth weight, gestational age, sex, and if they have been admitted to the NICU more than or less than 48 hours.

NICU admission alone confers 10 times increased risk of developing HL compared with the well-baby population. Based on the birth cohort for 2008–2020, the prevalence of congenital HL in our NICU population was 0.5%, significantly higher than in the well-baby population [35]. Others have found similar or greater levels of HL in their NICU population [2, 5]. If we consider only babies admitted for more than 48 hours then the prevalence rises to 1%, which could be a more realistic value since babies who are well but have suspected sepsis were included in NICU admission data during the data collection period.

### 4.1 UK and US risk factors

There are more very low birth weight infants surviving at the limits of viability, as new technologies and treatments continue to emerge. This constant evolution of therapeutic interventions and the characteristics of the NICU population have contributed to the limitations in using a set of risk factors to predict outcome.

Kraft (2014) confirmed that most neonatal indicators currently recommended by the JCIH are effective at identifying infants at increased risk of HL. They found NICU stay >5 days and

exposure to ototoxic medications are only associated with small gains in the number of infants correctly identified as at risk of HL [36]. UK risk factors which we found to be significantly different from matched controls despite being very specific, are not sensitive for HL so solely using these criteria, babies who go on to develop hearing loss would be missed. However, including time spent on the NICU (>48 hours) increases the number of babies who would be targeted, for follow-up hearing tests, to 92% (80% controls). Collating these data may provide useful information in countries without comprehensive screening programmes and help them target referral for high-risk infants. However, there is a difference between lower income countries in the proportion of HL due to preventable causes. This is most likely because of the higher incidence of infections, such as cCMV infections as well as fewer programs to support maternal and child health. It is thought that approximately one third of cases of preventable infant hearing loss in lower income countries have infectious causes such as rubella and meningitis [8].

## 4.2 Individual risk factors for hearing loss

The neonatal indicators we recorded which were statistically significantly different between the HL and the matched control group in our study were craniofacial anomalies, early-onset infection and syndromes associated with HL. We recorded a higher number of craniofacial abnormalities in our HL population compared with the matched group—many from cases with UHL. Others have found similar; Meyer et al. (1999) found craniofacial malformations, familial hearing disorders and neonatal bacterial infections were significant risk factors, whereas complications of prematurity were not independent risk factors [22], dysmorphic features, low APGAR scores at 1 minute, sepsis, meningitis, cerebral bleeding, and cerebral infarction were all associated with HL.

**4.2.1 Time spent on the NICU.** In this study infants spent nearly twice as long on the NICU if they had a HL (23 days versus 14 days for matched controls). It is likely to be a multi-factor causation where babies receiving the highest level of care are sickest and are potentially exposed to more pharmaceutical agents, noise, and respiratory support etc. The idea of accumulation of risk factors has gained merit more recently with another case-controlled study showing HL may be associated with the combination of neonatal indicators and pharmaceutical agents accumulated over time [7].

It has been shown that controlling for the impact of other risk indicators, NICU length of stay greater than 5 days and exposure to loop diuretics were not associated with an increased risk of congenital or delayed onset HL [36]. However, their study was not case controlled, and they didn't look at different levels of care. NICU length of stay has been identified as a risk factor in other studies but the utility of using a length of stay greater than 5 days has not [37]. The rationale for NICU stay >5 days in the 2007 JCIH position statement comes from unpublished data from the National Perinatal Information Network (NPIN). This NPIN data indicates that half of infants discharged from NICUs in 2005 were discharged before 5 days, and these infants were significantly less likely to have identified risk indicators. In the current study most babies with HL spent more than 5 days on the NICU but this was not statistically greater than matched controls.

**4.2.2 HIE.** Fitzgerald (2019) found hearing impairment is common (9.5%) in term infants who have undergone therapeutic hypothermia for moderate/severe HIE [38]. We were not able to verify different stages of HIE from the patient records but found no difference in the small number of cases (n = 4) with HL and HIE compared to matched controls who had no HL (n = 7).

**4.2.3 Bilirubin.** Although hyperbilirubinemia is extremely common among neonates and is usually mild and transient, historically it has led to bilirubin-induced neurologic damage.

Bilirubin-induced auditory impairment is thought to primarily influence brainstem nuclei and the auditory nerve, leading to auditory neuropathy spectrum disorder[39]. It is difficult to make a definitive statement about this link as only four babies with HL (all BHL) had severe unconjugated hyperbilirubinemia, and only one baby received an exchange transfusion. Notably no babies in the matched control group had severe unconjugated hyperbilirubinemia.

**4.2.4 Neonatal infection.** Congenital HL is associated with *Toxoplasma gondii*, Rubella virus, CMV, herpes simplex virus and *Treponema pallidum* infections [10]. We recorded few confirmed TORCH infections. However, we did find that any record of early-onset infection indicated a significant increase in risk of HL compared to matched controls. Sepsis is a common condition in NICU infants with poor effect on general outcome and health and a strong association between sepsis and HL has been documented [40]. Reporting of cCMV was low in this cohort, it is possible that some of the ten babies who did not have any UK/US risk factors or genetic anomalies had cCMV. A recent study has shown that the yield of targeted cCMV screening following newborn hearing screening failure was eight times higher than the estimated national birth prevalence of cCMV and those babies with clinically unsuspected cCMV disease had confirmed HL [41]. Future research could investigate if those children without any neonatal indicators or risk factors for HL have cCMV.

Bacterial meningitis is a known cause of HL and is recorded in the list of risk factors for UK and US committees. Deafness is the commonest serious complication of bacterial meningitis in childhood, approximately 10% of survivors are left with permanent sensorineural hearing loss [42, 43]. Fortunately, in the UK meningitis is an uncommon condition and therefore is not found in many studies evaluating risk factors. We recorded two cases (3%) of confirmed meningitis in our HL cohort and one in the control group in 12 years.

**4.2.5 Ototoxic drug use.** Aminoglycoside antibiotics are widely used because of their extreme effectiveness and broad-spectrum specificity toward organisms common in neonatal sepsis. Yet, there is a risk of HL following transient nephrotoxicity in as many as 20% of adult patients receiving aminoglycosides for extended periods of time [44]. The literature is conflicting about ototoxic drugs as a risk factor for HL in NICU graduates. One study found HL was not associated with gentamicin use [38]. Most (94%) of our HL cases had at least one dose of an aminoglycoside but this was not significantly different from matched controls (90%). Useful peak and trough levels of aminoglycoside medication and number of days medication were not routinely recorded on our database. If the low prevalence (0.2%) of the m.1555A>G variant in the UK population [45] is accurate for our study population, then fewer than one baby in our 12-year HL cohort would have this variant. Evaluation of point of care testing for the m.1555A>G variant to guide antibiotic prescribing to avoid aminoglycoside ototoxicity has been recently reported [46].

**4.2.6 Genetics.** The frequency of hearing loss-associated gene mutations has been seen to be higher in the NICU population (3%) and the hearing loss-associated gene mutations in the NICU, suggests this mutation may interact with perinatal high-risk factors [47].

In the group with UHL a third of patients had a pathogenic mutation (not necessarily causative) and in the group with BHL there were four patients (10%) which have a pathogenic mutation, causative of HL. Two cases had the most common pathogenic mutation—GJB2 variant autosomal recessive non-syndromic hearing loss and both these cases would not have been targeted if they had passed the NHSP. The GJB*2* gene is located on the DFNB1 locus (13q11-q12), codes for the connexin 26 protein, it presents in the human cochlea from the 22nd week of embryonic development [48]. One infant with BHL had Ehlers-Danlos syndrome and one infant with BHL had a 17 Mb loss on chromosome 18 which is associated with craniofacial anomalies amongst other developmental abnormalities.

National guidelines in the UK recommend but do not mandate genetic testing, chromosomal studies, and CGH microarray for cases of BHL but only in specific instances for cases of

UHL. Genetic testing should be available for all children with UHL, particularly since UHL can be a red flag for other multiple congenital anomalies which may not come to light until the children are much older [14].

### 4.3 Future studies

A multi-centre study would be required to provide both regional and ethnic diversity to the study to make the results more generalisable. This would also be important in lower-income countries as their disease demographic differs significantly. Ideally, efforts to identify such risk indicators would be improved by increasing integration of data collection efforts at a national level.

### 4.4 Limitations

The principal limitation of the current study was the maximum number of HL cases which could be identified. Sampling bias may have been an issue since we only selected HL cases which had been referred from NHSP and then confirmed to have a HL. Babies with UHL or mild BHL could be missed because of current UK NHSP policy. Nearly half of children with HL experience deterioration in hearing during childhood [49, 50], therefore, follow-up of acquired and progressive HL would be important to study to find factors which may mitigate against progression or target children who are susceptible. Since our study was retrospective, we could not collect any data that was not routinely documented. Several risk factors (e.g., blood glucose, creatine, or serum concentrations of gentamicin) which may be relevant were not included in this study for this reason. We also could not pre-plan testing of individual neonatal indicators, and we needed to test significance of multiple factors.

## 5. Conclusions

This study confirms in our population that many risk factors currently recommended are highly specific for identifying infants at risk of congenital HL but have low sensitivity. However, sensitivity cannot easily be improved in situations where HL would not be identified by UK or US risk factors, as no other neonatal indicators were exposed in the group who had no UK/US risk factors. NHSP is the gold standard for identifying HL in babies and relying solely on neonatal indicators for referral is an inferior option indicating NHSP needs to be developed in every country. Undiagnosed congenital CMV and genetic anomalies may play a significant role and needs further investigation. Data integration with a national health body would be required to clarify or uncover novel risk factors for congenital HL. Preventative measures (e.g., antenatal steroids) and investigating clinical biomarkers of infection and neonatal HL would be the next step in paediatric HL research.

## Supporting information

**S1 Checklist. Human participants research checklist.**
(DOCX)

## Author Contributions

**Conceptualization:** Sally K. Thornton, Karen R. Willis, Padraig T. Kitterick, Dulip S. Jayasinghe.

**Data curation:** Sally K. Thornton, Alice M. Yates, Karen R. Willis, Polly Scutt, Padraig T. Kitterick, Dulip S. Jayasinghe.

**Formal analysis:** Sally K. Thornton, Alice M. Yates, Polly Scutt, Padraig T. Kitterick, Dulip S. Jayasinghe.

**Funding acquisition:** Padraig T. Kitterick.

**Investigation:** Sally K. Thornton, Alice M. Yates, Abhijit Dixit, Dulip S. Jayasinghe.

**Methodology:** Sally K. Thornton, Padraig T. Kitterick, Dulip S. Jayasinghe.

**Project administration:** Sally K. Thornton, Alice M. Yates.

**Resources:** Sally K. Thornton, Derek J. Hoare, Karen R. Willis.

**Software:** Sally K. Thornton.

**Supervision:** Sally K. Thornton, Padraig T. Kitterick, Dulip S. Jayasinghe.

**Validation:** Sally K. Thornton, Dulip S. Jayasinghe.

**Visualization:** Sally K. Thornton.

**Writing – original draft:** Sally K. Thornton, Padraig T. Kitterick, Dulip S. Jayasinghe.

**Writing – review & editing:** Sally K. Thornton, Derek J. Hoare, Karen R. Willis, Dulip S. Jayasinghe.

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
