## [Decision Letter · Decision Letter 0]

18 Dec 2023

PONE-D-23-23156UK and US risk factors for hearing loss in neonatal intensive care unit infantsPLOS ONE

Dear Dr. Thornton,

Thank you for submitting your manuscript to PLOS ONE. After careful consideration, we feel that it has merit but does not fully meet PLOS ONE’s publication criteria as it currently stands. Therefore, we invite you to submit a revised version of the manuscript that addresses the points raised during the review process.

**The paper is well written but requires some **
**modification**

We look forward to receiving your revised manuscript.

Kind regards,

Kazumichi Fujioka

Academic Editor

PLOS ONE

Journal Requirements:

"Funding of this study was provided by the National Institute for Health and Care Research (NIHR) Nottingham Biomedical Research Centre (BRC). The funder had no role in the design or conduct of the study, or production of the manuscript. The views therefore expressed are those of the authors, and not necessarily those of the NIHR, the NHS, or the Department of Health and Social Care."

"No authors have competing interests"

Reviewers' comments:

Reviewer's Responses to Questions

**Comments to the Author**

1. Is the manuscript technically sound, and do the data support the conclusions?

Reviewer #1: Yes

Reviewer #2: Yes

Reviewer #3: Yes

2. Has the statistical analysis been performed appropriately and rigorously? 

Reviewer #1: Yes

Reviewer #2: Yes

Reviewer #3: Yes

3. Have the authors made all data underlying the findings in their manuscript fully available?

Reviewer #1: Yes

Reviewer #2: Yes

Reviewer #3: Yes

4. Is the manuscript presented in an intelligible fashion and written in standard English?

Reviewer #1: Yes

Reviewer #2: Yes

Reviewer #3: Yes

5. Review Comments to the Author

Reviewer #1: Thanks

this paper address compare the risk factors UK and US risk factors for hearing loss in neonatal

intensive care unit infants, paper well done

US and UK risk factors both insufficient to make risk factor with high sensitivity and specificity, can the authors make new Table of combined high risk factors noticed to make new recommendation that if we used the new table the sensitivity and specificity to detect hearing loss will increased.

Reviewer #2: This study seeks to identify factors that can contribute to the identification of hearing loss in NICU babies. The main finding of high specificity with UK and high sensitivity with US criteria is interesting in that both criteria may need to be employed to help in this assessment.

There are several statements that need clarification to improve readability and comprehension of the findings:

p4 line 88: JCIH guidelines recommend diagnostic identification by 3 mo and intervention by 6 months of age in cases of failed hearing screening or higher risk categories

p5 lines 119-122: you state there was a paucity of tympanometry data and therefore could not determine conductive or sensorineural hearing loss. ABR testing would be a far better method to determine specific type and degree of hearing loss, tympanometry could be beneficial to rule out temporary conductive hearing loss due to middle ear pathology but does not differentiate between types of permanent hearing losses.

The study does not state how long these children were followed and if the hearing loss changed over time or not. (line 131 describing the matched cohort)

please describe the guidelines used for identification of CMV (line 151)

It is somewhat surprising that out of 14,000 children only 71 were found to have hearing loss - this is similar to that of the general population but seems low for NICU babies. Can you explain this finding?

Was imaging obtained for the 10 babies who did not meet the risk factors and if so please describe that ( 1 was listed as EVA).

Limitations are well described

Reviewer #3: This single center study is a matched case control analysis of patients confirmed to have hearing loss to evaluate the sensitivity and specificity of risk factors from hearing screening guidelines in the United States and the United Kingdom. Despite acknowledged limitations due to small sample size, this work provides a useful comparison of these guidelines in the context of a UK based study cohort.

Introduction:

This is a helpful overview of what is known regarding hearing loss in newborns. Given that this work is focused on a comparison of the utility of US vs UK hearing loss guidelines, I think it would be beneficial to bring reference to those guidelines into the introduction and provide an overview of the similarities and differences. Many readers will be familiar with one but not the other set of guidelines.

Materials and Methods

Line 115 – the acronym BSA is utilized but not defined. I do not see the BSA guidelines referenced in this initial use in the manuscript. Can you add a reference to these guidelines?

Line 143 (Table 1) – this may be a formatting issue, but the numbering of risk factors on the UK side of the table begins at #3 and I do not see a #1 or #2 in the table. The US risk factors start at #7 rather than beginning at #1. The last three US risk factors are bullet points rather than numbered.

Results

No major concerns about the results as provided.

The data provided in table 4 seems more effectively summarized in a few sentences and perhaps does not need a full table given that only 2 of the provided patients had documented genetic abnormalities.

Discussion

The discussion provides a detailed review of several risk factors for hearing loss including one or more paragraphs on NICU length of stay, bilirubin, neonatal infection, ototoxic drug use, and genetics. The discussion could be made more concise if these dives into individual risk factors are restricted to only those that were found to be most significant in this cohort with a link back to what is generally known about these risk factors. As an example, ototoxic drug use is not significantly different between cases and controls in the cohort, so I don’t believe there is reason to devote a paragraph to this topic in this discussion.

6. PLOS authors have the option to publish the peer review history of their article (what does this mean?). If published, this will include your full peer review and any attached files.

Reviewer #1: **Yes: **Ramadan A Mahmoud

Reviewer #2: No

Reviewer #3: No

---

## [Author Response · Author response to Decision Letter 0]

3 Feb 2024

Response to Reviewers

Reviewer #1: Thanks

this paper address compare the risk factors UK and US risk factors for hearing loss in neonatal

intensive care unit infants, paper well done

US and UK risk factors both insufficient to make risk factor with high sensitivity and specificity, can the authors make new Table of combined high risk factors noticed to make new recommendation that if we used the new table the sensitivity and specificity to detect hearing loss will increased.

RESPONSE: That is a great idea but unfortunately it is not possible as there is no perfect combination of individual risk factors for high specificity and sensitivity. It is a very varied combination of different risk factors/neonatal indicators which impacts sensitivity and specificity, and these are not the same for all children with hearing loss but vary across the group. In a very large multi-centred group this may be possible.

Reviewer #2: 

Comment: This study seeks to identify factors that can contribute to the identification of hearing loss in NICU babies. The main finding of high specificity with UK and high sensitivity with US criteria is interesting in that both criteria may need to be employed to help in this assessment.

There are several statements that need clarification to improve readability and comprehension of the findings:

1) p4 line 88: JCIH guidelines recommend diagnostic identification by 3 mo and intervention by 6 months of age in cases of failed hearing screening or higher risk categories

RESPONSE: This sentence had been revised to read, “Regardless of their hearing screening result, many NICU babies meet at-risk criteria for referral and additional hearing testing at 7-9 months according to US (Joint committee on infant hearing [JCIH]) and UK (National Institute for Health and Care Excellence NHSP) guidelines.”

2) p5 lines 119-122: you state there was a paucity of tympanometry data and therefore could not determine conductive or sensorineural hearing loss. ABR testing would be a far better method to determine specific type and degree of hearing loss, tympanometry could be beneficial to rule out temporary conductive hearing loss due to middle ear pathology but does not differentiate between types of permanent hearing losses.

Thank you, we have been able to collect the air and bone conduction diagnostic ABR data so we are now able to include the types of hearing loss for this cohort.

RESPONSE: This sentence has now been reworded to, ‘The hearing data extracted from air and bone conduction tests (diagnostic ABR) indicated that 59 children had sensorineural hearing loss (17 with auditory neuropathy spectrum disorder), 7 had conductive hearing loss and 5 had mixed hearing loss.’

Comment: The study does not state how long these children were followed and if the hearing loss changed over time or not. (line 131 describing the matched cohort)

RESPONSE: In this retrospective cohort study, we are only studying confirmed permanent congenital hearing loss, acquired hearing loss cases are not included in this cohort and the impact of neonatal indicators/risk factors on whether the child’s hearing progressively declines is outside the remit of this study. Data from the children’s notes were taken from birth until data collection (2022-2023). We identified cases with permanent hearing loss from the NHSP and also their diagnostic ABR as a baby, however their degree of hearing loss was taken from their most recent hearing test results as documented in the methods. 

Comment: please describe the guidelines used for identification of CMV (line 151)

RESPONSE: We have now included the online reference for the local guideline.

Comment: It is somewhat surprising that out of 14,000 children only 71 were found to have hearing loss - this is similar to that of the general population but seems low for NICU babies. Can you explain this finding?

RESPONSE: The reviewer is correct and we may be missing cases of UHL and mild BHL because of UK NHSP policy (see selection bias in the limitations section of the discussion). 

Line 360 (Limitations in the discussion): ‘Babies with UHL or mild BHL could be missed because of current UK NHSP policy.’

In addition, as noted in the discussion, the percentage of children with hearing loss could be closer to 1% since all babies with suspected sepsis were admitted during the time of data collection to the NICU so there is inflated population numbers which reduces prevalence.

Comment: Was imaging obtained for the 10 babies who did not meet the risk factors and if so please describe that (1 was listed as EVA).

RESPONSE: No there was a paucity of imaging data (n=2) that were available to us and is described in text. We have now added in the sentence. ‘Imaging data were only available for two of these children where one had a normal MRI and the other one had dilatation of vestibular aqueducts.’

Reviewer #3: 

Comment: Given that this work is focused on a comparison of the utility of US vs UK hearing loss guidelines. I think it would be beneficial to bring reference to those guidelines into the introduction and provide an overview of the similarities and differences. Many readers will be familiar with one but not the other set of guidelines.

RESPONSE: Agreed - we have moved table 1 in the Methods to the Introduction which summarises the UK and US risk factors. Also added the sentence above, ‘Table 1. summarises current UK and US risk factors for hearing loss.’

Comment: Line 115 – the acronym BSA is utilized but not defined. I do not see the BSA guidelines referenced in this initial use in the manuscript. Can you add a reference to these guidelines?

RESPONSE: Apologies this was an omission. Reference for BSA (British Society of Audiology) is now added in (www.thebsa.org.uk/) and the acronym is spelt out.

Comment: Line 143 (Table 1) – this may be a formatting issue, but the numbering of risk factors on the UK side of the table begins at #3 and I do not see a #1 or #2 in the table. The US risk factors start at #7 rather than beginning at #1. The last three US risk factors are bullet points rather than numbered.

RESPONSE: We have reformatted Table 1.

Comment: The data provided in table 4 seems more effectively summarized in a few sentences and perhaps does not need a full table given that only 2 of the provided patients had documented genetic abnormalities.

RESPONSE: The authors agree and I have removed table 4 and have added in additional text as follows: ‘Imaging data was only available for two of these children where one had a normal MRI and the other one had dilatation of vestibular aqueducts.’

Comment: The discussion provides a detailed review of several risk factors for hearing loss including one or more paragraphs on NICU length of stay, bilirubin, neonatal infection, ototoxic drug use, and genetics. The discussion could be made more concise if these dives into individual risk factors are restricted to only those that were found to be most significant in this cohort with a link back to what is generally known about these risk factors. As an example, ototoxic drug use is not significantly different between cases and controls in the cohort, so I don’t believe there is reason to devote a paragraph to this topic in this discussion.

RESPONSE: We appreciate the suggestion to focus the discussion more on those variables where there was a significant difference between groups. However, given the sample size, we consider it more relevant to retain our broader discussions of risk factors in terms of their comparison to the matched controls despite statistical significance which will be very much influenced by our small sample. These are all risk factors we consider need to be pursued in further and larger studies. 

---

## [Decision Letter · Decision Letter 1]

22 Apr 2024

UK and US risk factors for hearing loss in neonatal intensive care unit infants

PONE-D-23-23156R1

Dear Dr. Thornton,

We’re pleased to inform you that your manuscript has been judged scientifically suitable for publication and will be formally accepted for publication once it meets all outstanding technical requirements.

Kind regards,

Kazumichi Fujioka

Academic Editor

PLOS ONE

Additional Editor Comments (optional):

Reviewers' comments:

Reviewer's Responses to Questions

**Comments to the Author**

1. If the authors have adequately addressed your comments raised in a previous round of review and you feel that this manuscript is now acceptable for publication, you may indicate that here to bypass the “Comments to the Author” section, enter your conflict of interest statement in the “Confidential to Editor” section, and submit your "Accept" recommendation.

Reviewer #1: All comments have been addressed

Reviewer #2: All comments have been addressed

2. Is the manuscript technically sound, and do the data support the conclusions?

Reviewer #1: Yes

Reviewer #2: Yes

3. Has the statistical analysis been performed appropriately and rigorously? 

Reviewer #1: Yes

Reviewer #2: Yes

4. Have the authors made all data underlying the findings in their manuscript fully available?

Reviewer #1: Yes

Reviewer #2: Yes

5. Is the manuscript presented in an intelligible fashion and written in standard English?

Reviewer #1: Yes

Reviewer #2: Yes

6. Review Comments to the Author

Reviewer #1: still ask about the most signs funded by authors may had higher sensitivity in screening Newborn for hearing test from both US UK

Reviewer #2: comments adequately cover the concerns. Thank you for the revisions. the modified table and the addition of the ABR information makes the study more understandable.

7. PLOS authors have the option to publish the peer review history of their article (what does this mean?). If published, this will include your full peer review and any attached files.

Reviewer #1: **Yes: **Ramadan A Mahmoud

Reviewer #2: No

---

## [Editor Report · Acceptance letter]

19 Jun 2024

PONE-D-23-23156R1 

PLOS ONE

Dear Dr. Thornton, 

I'm pleased to inform you that your manuscript has been deemed suitable for publication in PLOS ONE. Congratulations! Your manuscript is now being handed over to our production team.

Kind regards, 

on behalf of

Dr. Kazumichi Fujioka 

Academic Editor

PLOS ONE